# Leptin Downregulates Angulin-1 in Active Crohn’s Disease via STAT3

**DOI:** 10.3390/ijms21217824

**Published:** 2020-10-22

**Authors:** Jia-Chen E. Hu, Christian Bojarski, Federica Branchi, Michael Fromm, Susanne M. Krug

**Affiliations:** 1Institute of Clinical Physiology/Nutritional Medicine, Charité—Universitätsmedizin Berlin, Campus Benjamin Franklin, 12203 Berlin, Germany; 2Department of Gastroenterology, Rheumatology and Infectious Diseases, Charité—Universitätsmedizin Berlin, Campus Benjamin Franklin, 12203 Berlin, Germany

**Keywords:** leptin, Crohn’s disease, tight junction, angulin-1

## Abstract

Crohn’s disease (CD) has an altered intestinal barrier function, yet the underlying mechanisms remain to be disclosed. The tricellular tight junction protein tricellulin is involved in the maintenance of the paracellular macromolecule barrier and features an unchanged expression level in CD but a shifted localization. As angulins are known to regulate the localization of tricellulin, we hypothesized the involvement of angulins in CD. Using human biopsies, we found angulin-1 was downregulated in active CD compared with both controls and CD in remission. In T84 and Caco-2 monolayers, leptin, a cytokine secreted by fat tissue and affected in CD, decreased angulin-1 expression. This effect was completely blocked by STAT3 inhibitors, Stattic and WP1066, but only partially by JAK2 inhibitor AG490. The effect of leptin was also seen at a functional level as we observed in Caco-2 cells an increased permeability for FITC-dextran 4 kDa indicating an impaired barrier against macromolecule uptake. In conclusion, we were able to show that in active CD angulin-1 expression is downregulated, which leads to increased macromolecule permeability and is inducible by leptin via STAT3. This suggests that angulin-1 and leptin secretion are potential targets for intervention in CD to restore the impaired intestinal barrier.

## 1. Introduction

Crohn’s disease (CD) is a chronic idiopathic relapsing and remitting gastrointestinal condition with a climbing prevalence in western countries and an increasing incidence in developing regions [1,2]. Besides genetic predisposition, environmental influence and microbiota, the pathogenesis of CD might also result from miscommunication between intestinal epithelial cells and the immune system [3]. Cytokines play a compelling role in balancing immune function, but also could exert pathological effects on the immune system when excessively produced. Leptin is a special kind of cytokine secreted largely by adipose tissue. Hypertrophic and hyperplastic adipocytes could lead to cell apoptosis, hypoxia, macrophages infiltration, and proinflammatory cytokines releasement (i.e., TNF-α, IL-1β and IL-6) [4]. In CD, a characteristic thickened mesenteric fat tissue adjacent to inflamed intestinal segments was described [5]. This “creeping fat” could be an important source of additional cytokine release. Under inflammatory conditions, an increasing amount of leptin could lead to the damage of the epithelial wall and the infiltration of neutrophils [6,7,8].

The tight junction (TJ) acts as a critical part in maintaining the intactness of the intestinal barrier and is considered a determinant of the paracellular transport. On one hand, it behaves as a “fence” that prevents movement of membrane constituents between apical and basolateral cell membranes [9]. On the other hand, the TJ has a “gate function” that controls the passage of water, ions, small water-soluble molecules, and macromolecules through the intercellular space [10]. The structure of the TJ comprises two patterns, the bicellular TJ (bTJ) formed by a belt-like meshwork of strands between two epithelial cells [11] and the tricellular TJ (tTJ) which is situated in the region where three cells meet and consists of vertically extended bTJs, forming a central tube [12]. Due to the space left in the central tube, the tTJ is assumed to be a weak region for the total paracellular barrier and by this a potential route for transepithelial fluxes. Tricellulin and the angulin family are the major protein components of tTJ. Tricellulin is essential for maintaining an intact tTJ assembly as well as its barrier function, especially that for macromolecules [13,14].

Why this is essential deserves some explanation. Under normal conditions, the intestinal epithelial barrier prevents significant uptake of luminal antigens into the lamina propria. However, if the tTJ barrier is opened for the passage of large molecules, luminal antigens can enter the lamina propria. There, they stimulate local immune cells to develop a proinflammatory response by releasing chemokines which exacerbate the intestinal inflammatory process in CD [3,15]. 

The angulin family comprises three members featuring a common ability to recruit tricellulin to the tTJ [16]. Angulin-1, also named lipolysis-stimulated lipoprotein receptor (LSR), was initially recognized as a rate-limiting step in the progress of lipid clearance [17] and later identified to have a primary localization at the tTJ [18]. Angulin-1 could recruit tricellulin to the tTJ, and knockdown of angulin-1 leads to a decreased transepithelial resistance (TER) and an increased permeability to fluorescein and macromolecules up to 40 kDa [18], indicating that angulin-1—probably indirectly acting by the removal of tricellulin from the tTJ—was also essential for sustaining the epithelial barrier. To date, angulin-1 has been linked to lipid metabolic abnormality [19,20,21], Alzheimer’s disease [22], and various cancers [23,24,25,26,27,28,29,30]. Akin to tricellulin, angulin-1 is also targeted by bacteria in order to breach the intestinal barrier [31,32]. In vitro cell culture experiments also showed leptin downregulating angulin-1 at pathological concentrations [27,33]. In all epithelia with the expression of tricellulin, at least one angulin localizes at the tTJ and in colon the major role falls on angulin-1 [16].

Dysregulation of TJ proteins could cause or could be caused by corresponding diseases due to altered paracellular passage of water, solutes, and macromolecules [34]. In CD, TJs are altered in ultrastructure as well as protein expression and localization [35]. Expression of tricellulin was found to be shifted from depths of crypts to surface epithelium in CD while its expression was unaltered [15]. Although the regulatory mechanisms behind this shifting and the underlying components are unexplored, a reasonable hypothesis would be that the displacement of tricellulin might result from elements responsible for its correct localization.

In this study, we focused on analyzing the expression of tTJ proteins in CD and found angulin-1 to be downregulated. Then, we also aimed to discover the affecting factors behind this and found out that leptin was able to cause the expressional alteration of angulin-1 and that the involved mediator was STAT3.

## 2. Results

### 2.1. Patients Features

In total, 19 CD patients and 24 patients without intestinal diseases who underwent colonoscopy were enrolled for the study between 2017 and 2020 (Table 1). Simple endoscopic score for CD (SES-CD) is a simplified scoring system that is commonly used to evaluate the endoscopic presentation of CD patients, the items of which includes mucosal ulcers, surface involved by CD or ulceration, and the presence of narrowing [36].

### 2.2. Expression of Angulins in Intestinal Biopsies

Regarding mRNA levels, angulin-1 showed a downregulation in active CD compared with healthy controls (Figure 1a, * *p* < 0.05,) and a recovered expression in remission CD compared with CD-active patients (Figure 1a, * *p* < 0.05,). Despite not being significant, the similar pattern could be found in angulin-2 expression (Figure 1b). The expression of angulin-3 remained unchanged in CD (Figure 1c). As controls, expression of claudin-2, -4 and tricellulin were checked. Claudin-2 (Cldn-2) was upregulated as already known in active CD compared to healthy controls (Figure 1e, ** *p* < 0.01,) as well as to CD in remission (Figure 1e, * *p* < 0.05). The expression of tricellulin and claudin-4 (Cldn-4) remained unchanged in CD (Figure 1d,f).

Regarding protein expression levels, angulin-1 in formalin-fixed paraffin-embedded (FFPE) intestinal biopsies was decreased in CD when compared to healthy controls (Figure 2a,b, * *p* < 0.05). Since epithelial cells may not be homogeneously distributed within each paraffin-embedded section, the protein expression from freshly taken biopsies was then also analyzed to avoid sample selection bias. Protein extracted from the whole colonic biopsy which did not undergo the fixation or embedding procedure confirmed a reduced expression of angulin-1 in active CD compared with Ctrl as well as remission CD (Figure 2c,d, ** *p* < 0.01). As controls, we also analyzed the protein expression of tricellulin and Cldn-4 and found them to be unaltered as previously demonstrated [15] (Appendix A).

### 2.3. Cytokine Effects of Angulins in Human Intestinal Epithelial Cell Lines

In order to figure out the potential responsible cytokines of the downregulated angulin-1 in CD, intestinal epithelial cell lines T84 and Caco-2 were cultured for the treatment of twelve cytokines which were reported in the literature to possess proinflammatory effects (TNF-α, IFN-γ, leptin, IL-1β, IL-6, IL-12, IL-17A, IL-17F, IL-21, IL-22, IL-23 and IL-33).

For T84 cells, leptin was the only cytokine that downregulated angulin-1 (to 81.11 ± 2.07% of untreated Ctrl after 48 h), while all the other cytokines investigated showed no effect (Appendix A, *** *p* < 0.001, *n* = 12). There was no change of TER after leptin treatment (Appendix Ac).

For Caco-2 cells, angulin-1 protein expression was also only decreased by leptin to 82.38 ± 1.03% (Appendix Ac, *** *p* < 0.001, *n* = 12) and was increased by TNF-α or IL-17F (Appendix Aa, ** *p* < 0.01, *n* = 12; Appendix Ah, * *p* < 0.05, *n* = 9). The TER value also did not change after leptin treatment (Appendix Ac).

To further explore the effect of leptin, T84 and Caco-2 cells were incubated for different periods (24, 48, and 96 h) to compare the protein expression course of angulin-1 and tricellulin. In T84 cells, compared to the effect at 48 h described above, angulin-1 protein expression after 96 h of leptin treatment was further reduced to 58.66 ± 2.62% of untreated Ctrl (Figure 3a, *** *p* < 0.001). Tricellulin protein expression was stable at all three time points (Figure 3b).

In Caco-2 cells, the results were similar: angulin-1 protein expression after 96 h of leptin treatment was downregulated to 67.71 ± 1.68% compared to 24 h (Figure 4a, *** *p* < 0.001, *n* = 6) as well as 48 h (Figure 4a, *** *p* < 0.001, *n* = 12), and the expression level of angulin-1 at 48 h was also lower than at 24 h (Figure 4a, *** *p* < 0.001). Additionally, in Caco-2 cells there was no difference in tricellulin expression at all time points tested (Figure 4b).

### 2.4. Barrier Function of T84 and Caco-2 Cells Treated with Leptin

To explore whether the downregulation of angulin-1 by leptin could affect the macromolecule barrier function, permeability for the macromolecule marker FITC-dextran 4 kDa (FD4) was measured in T84 and Caco-2 cells after leptin treatment. TER values were not different with or without leptin treatment in both cell lines (Figure 5a,c). Regarding FD4 permeability, no significant change was observed in T84 cells (Figure 5b), but values in Caco-2 cells nearly doubled (Figure 5d, * *p* < 0.05).

### 2.5. Tricellulin Localization after Leptin Treatment

To determine whether the treatment with leptin could lead to a spatial shift of tricellulin, we performed immunofluorescent staining to exhibit the localization. Tricellulin was not affected by leptin treatment as its localization at tTJs remained unchanged (Figure 6).

### 2.6. Signaling Pathway of Leptin

To investigate the signal transduction of leptin-induced angulin-1 downregulation, several inhibitors of known signaling pathways (e.g., JAK2 [37], STAT3 [27], ERK1/2 [38], and PI3K [39]) of leptin were added before treatment. STAT3 inhibitors, Stattic as well as WP1066, were able to block the effect of leptin in both T84 (Figure 7a, *** *p* < 0.001, *n* = 6) and Caco-2 cells (Figure 7b, *** *p* < 0.001, *n* = 6), while in both cell lines this effect could only be partially inhibited after the blockage of the assumed upstreaming JAK2 pathway (Figure 7a,b, * *p* < 0.05, *n* = 6).

To further elucidate the regulation of STAT3, a phosphorylation protein analysis was carried out and showed an increase in phosphorylation, becoming significant after 30 min (T84 cells, Figure 8a, ** *p* < 0.01, *n* = 7; Caco-2 cells, Figure 8b, * *p* < 0.05, *n* = 6) and reaching a peak at 60 min after leptin treatment (T84 cells, Figure 8a, *** *p* < 0.001, *n* = 11; Caco-2 cells, Figure 8b, *** *p* < 0.001, *n* = 9).

Next, STAT3 inhibitors, Stattic and WP1066, and the JAK2 inhibitor AG490, which showed an inhibitory effect to leptin treatment, were applied for 1 h before leptin treatment. Similar to the effects seen on angulin-1 expression in both cell lines, Stattic as well as WP1066 inhibited the phosphorylation of STAT3 (Figure 9a,b, *** *p* < 0.001, *n* = 6–8), whereas AG490 only showed a partial effect (Figure 9a, ** *p* < 0.01, *n* = 8; Figure 9b, * *p* < 0.05, *n* = 6).

## 3. Discussion

### 3.1. The Involvement of Angulin-1 in CD

The tTJ is a special arrangement of TJ strands at the contacts of where three or more cells meet. The structure of the tTJ forms a vertically extended ladder-like meshwork and has been assumed to be a weak point of paracellular barriers [11]. Tricellulin and the angulin family are the two major components of tTJs and have been shown to be involved in maintaining the barrier of the tTJ. While angulins are responsible for the correct localization of tricellulin, this protein tightens the tTJ against macromolecule passage [14,15,40].

Despite the difference of the principal angulin expression in different tissues, angulin-1 is the major one in the base of crypts, while the expression of angulin-2 fills the vacancy of the upper part of crypts [16].

In a previous study analyzing human sigmoid colon samples it was found that tricellulin retained the same protein expression level in CD and control patients, yet further investigation of tricellulin localization revealed a mild decrease within crypts but an increase in surface epithelium [15]. This shifted localization suggested that the regulation of tricellulin in CD might involve regulators, and that this alteration of tTJs might be more sophisticated. For this purpose, the expression of angulins was analyzed in CD. We show here that angulin-1 was downregulated specifically in active CD compared to controls as well as CD patients in remission. Angulin-2 and -3 were unaltered; however, due to the insufficient quality of the available antibodies against angulin-2 and -3, these two proteins could only be analyzed at the mRNA level.

### 3.2. Leptin Affecting the Expression of Angulin-1

To elucidate the mechanisms behind the downregulation of angulin-1 in CD, leptin, among the total twelve cytokines, which have been reported to be mainly involved in CD, was found to be the only one leading to a decrease in angulin-1 in the intestinal cell lines T84 and Caco-2. The involvement of leptin was consistent with its enhanced effects on Th1 as well as Th17 cytokines and also with the hypertrophic MAT in CD [41].

Analyzing the functional effect of the leptin treatment and thus angulin-1 downregulation, we observed that permeability for FD4 was increased in Caco-2 cells but not in T84 cells. This difference might be explained by the different permeability for FD4 in these two cell lines. Untreated Caco-2 cells have a lower permeability for FD4 compared to T84 cells. After treatment, the permeability increased to levels comparable of that of the untreated T84, which could suggest that this was already the maximum increase to be expected in such conditions and concentration, so that the difference is detectable in CaCo-2 cells, but not in T84. In a tissue similar to the intestinal crypt a well-balanced barrier is essential, so the impaired tTJ may be of bigger impact for the total macromolecular barrier than data from intestinal monolayers would suggest.

We did not observe localization changes of tricellulin in both T84 and Caco-2 cells using confocal laser scanning microscopy. This seems to be contradictory to the previous finding that the knockdown of angulin-1 resulted in a deviation of tricellulin from tTJs [18]. However, this experiment was performed under the circumstance of an extreme knockdown of angulin-1 while, in our present study, leptin only led to a downregulation of 40–50% of angulin-1. Furthermore, a quantification of the tricellular signals of tricellulin stainings was not performed as for this a strict calibration was needed. Our study might further indicate that a halved expression of angulin-1 is still sufficient to hold tricellulin in position, but the barrier becomes vulnerable compared to full angulin-1 expression as we did demonstrate an increased permeability for FD4 in Caco-2 cells. Interaction of tricellulin with other tricellulin molecules and angulin-1 could also not be visualized directly. This interaction might be reduced and thus lead to the observed weakening of the barrier. In addition, we did not analyze effects of the other angulins that could also keep tricellulin in place but were reported to be insufficient in keeping the barrier function [16], as the antibodies for these proteins are not of good quality and mRNA expression would not give any information about the actual interaction and localization of the proteins.

### 3.3. Leptin-Regulated Downregulation of Angulin-1 via STAT3 Pathway

Leptin usually signals via its three types of receptor isoforms: (i) the long isoform which contains an extracellular binding domain, a single transmembrane domain and an intracellular signaling domain [42], (ii) the short isoform which has three variants in humans and which lacks part of the cytoplasmic region, and (iii) the soluble isoform which is a cleavage product of the long isoform for binding circulating leptin [43].

Leptin receptors (LRs) are reported to lack intrinsic kinase activity; instead, the signal transduction has to process through the binding of Janus kinase (JAK) 2 and the phosphorylation of tyrosine sites downstream of the LR/JAK2 complex [37]. Therefore, above all sorts of LRs, the long-form LR is assumed to be most important in transducing leptin signals due to the fact that besides the binding motif for JAK2, it is the only isoform containing three tyrosine phosphorylation sites (Tyr986, Tyr1079 and Tyr1141 in humans [43] or Tyr985, Tyr1077 and Tyr1138 in rodents [44]).

Tyr985 binds to the Src-homology-2 domain protein (SHP-2) activating the ERK signaling pathway which mediates the regulation of c-fos message [38]. Tyr1077 activates STAT5 signaling acting on the reproduction effect of leptin [44]. Tyr1138 not only has the ability to phosphorylate STAT5, but also mediates STAT3 regulation, which could be subsequently inhibited by the interaction between the suppressor of cytokine signaling 3 (SOCS3) and Tyr985 [45] or JAK2 itself [46]. In addition, the involvement of PI3K was exhibited during the regulation of energy consumption in the central nerve system [39]. Leptin could also induce an anti-apoptotic effect through PI3K as well as mitogen-activated protein kinase (MAPK) pathways [47], which consequently enhanced Th1 and Th17 responses and promoted a proinflammatory effect [48,49,50].

Using different inhibitors, we show that the downregulation of angulin-1 induced by leptin was blocked by pre-treatment with AG490, Stattic or WP1066. Furthermore, phosphorylation experiments also revealed that STAT3 was phosphorylated after leptin treatment. These results confirmed the involvement of JAK2 and STAT3, but the incomplete blockage of AG490 suggested that leptin-activating STAT3 might not necessarily signal via JAK2.

### 3.4. Concluding Remarks

Leptin-1 is an important factor in fat metabolism, and angulin-1 was initially found to be involved in lipid clearance [17]. In line with our observed permeability change, a recent study reported that administration of lipid micelles in Caco-2 cells was able to cause an enhanced passage of macromolecules (>4 kDa). Lipid micelles decreased tricellulin content within tTJs, which also links tricellulin and the tTJ to lipid metabolism, presumably due to angulin-1 as a regulator [51].

One characteristic of CD is the appearance of “creeping fat”, a thickened mesenteric fat tissue located close to the inflamed parts of the intestine [5], pointing towards disturbed fat metabolism. Importantly, creeping fat is a source of leptin [6,7,8].

Here we show that angulin-1 is affected in the process of active CD, which can be driven by leptin via the STAT3 signaling pathway.

As an invasion of macromolecules (e.g., pathogens) might be an important reason in the relapse of CD, restoring the expression of angulin-1 might help tighten the intestinal barrier and break the loop of impaired barrier and increased uptake of pathogens. Therefore, targeting leptin as well as angulin-1 regulation might be potential approaches for new CD treatments.

## 4. Materials and Methods

### 4.1. Patients and Study Criteria

CD patients and patients without intestinal diseases visiting the endoscopy center of the Department of Gastroenterology, Rheumatology and Infectious Diseases, Charité—Universitätsmedizin were enrolled to the study with prior consent. The diagnosis of CD was based on the standard criteria [52] and the endoscopic activity was evaluated using simple endoscopic score for CD (SES-CD) [36]. The study was approved by the local ethics committee (No. EA4/015/13).

Exclusion criteria were: age below 18 or above 80 years, pregnancy, presence of other major diseases (neoplastic diseases, other immunological diseases and chronic inflammatory diseases), biotic treatment, presence of fistula or perforation, need of surgery, lack of consent to the study.

Healthy controls were patients free of gastrointestinal diseases (e.g., IBD, irritable bowel syndrome, diarrhea, preceding gastrointestinal surgery) and other major conditions described above.

### 4.2. Cell Lines

Two human intestinal epithelial cell lines, T84 cells (ATCC^®^ CCL-248™), and Caco-2 cells (ATCC^®^ HTB-37™) were cultured at 37 °C in a 5% CO_2_ air atmosphere. For cultivating, 25 cm^2^ culture flasks were used and complete growth mediums for these three cell lines are listed in Table 2. Medium was renewed every two to three days.

For experiments, 4 × 10^5^ cells were seeded onto 3 μm-pore-size (T84 cells, effective area 0.6 cm^2^, Millipore, MA, USA) or 0.4 μm-pore-size (Caco-2 cells, effective area 0.6 cm^2^, Millipore, MA, USA) cell culture inserts. Confluent and well-differentiated cell monolayers were used 8 days (T84), and 14 days (Caco-2) after seeding.

### 4.3. Cytokines and Inhibitors Experiments

For cytokine treatments, cells were incubated with the cytokines listed in Table 3 for 24 (TNFα) or 48 h (all the other cytokines). For inhibitor experiments, a 1 h pre-treatment was performed prior to leptin treatment at a listed concentration. For 96 h of leptin treatment with or without inhibitors, the medium was renewed after 48 h along with the same concentration of leptin and corresponding inhibitors.

### 4.4. Western Blotting

For isolation of protein from formalin-fixed paraffin-embedded (FFPE) sections, a Qproteome FFPE Tissue Kit (QIAGEN, Hilden, Germany) was used. FFPE Intestinal biopsies were acquired from sample stocks in the Institute for protein preparation. The protein extraction process was carried out according to the manufacturer’s instruction. RNA and protein were extracted in parallel using a NucleoSpin RNA/Protein kit (Macherey-Nagel, Düren, Germany). The concentration of protein was determined using a bicinchoninic acid (BCA) protein assay. Total protein from cell culture, phosphorylation assay, and Western blotting were performed as previously described [15]. The primary antibodies applied in the study were rabbit anti-angulin-1 (1:3000, Sigma-Aldrich, Schnelldorf, Germany), rabbit anti-tricellulin (1:2000, Invitrogen, Karlsruhe, Germany), rabbit anti-Cldn-2 (1:1000, Invitrogen, Karlsruhe, Germany), mouse anti-Cldn-4 (1:1000, Invitrogen, Karlsruhe, Germany), mouse anti-β-actin (1:10,000, Invitrogen, Karlsruhe, Germany), rabbit anti-STAT3 and rabbit anti-phospho-STAT3 (1:1000, Cell Signaling Technology). For secondary antibodies, peroxidase-conjugated goat anti-rabbit or -mouse IgG (Jackson ImmunoResearch, Ely, UK) was used. Image detection was performed with incubation with a SuperSignal West Pico Plus Stable Peroxide solution (Thermo Fisher, Mannheim, Germany) and exposure in a Fusion FX7 (Vilber Lourmat, Eberhardzell, Germany). Densitometric analysis was performed using Multi Gauge V2.3 software (FujiFilm, Düsseldorf, Germany). The expression of angulin-1, tricellulin, Cldn-2 and Cldn-4 was quantified after normalization of the respective band intensities using β-actin.

### 4.5. RNA Isolation, Reverse Transcription, and Quantitative Real-Time PCR (qRT-PCR)

The isolation of RNA was described previously [15]. Reverse transcription was performed using a High Capacity cDNA Reverse Transcription kit (Thermo Fisher, Mannheim, Germany). Angulin-1 (Hs01076323_m1), angulin-2 (Hs01111433_m1), angulin-3 (Hs01025498_m1), tricellulin (Hs00930631_m1), Cldn-2 (Hs00252666_s1), Cldn-4 (Hs00533616_s1), and GAPDH (Hs02786624_g1) TaqMan^®^ probes were used in qRT-PCR. A relative method according to the 2^−ΔΔCT^ method was applied for expression analysis.

### 4.6. Immunofluorescent Staining

After leptin treatment, the cell monolayers were fixed with 2% PFA, quenched with 25 mM glycine, permeabilized using 0.5% Triton X-100, and blocked with 5% goat serum. The cells were stained with the following primary antibodies: rabbit anti-angulin-1 (1:1500, Sigma-Aldrich, Schnelldorf, Germany), rabbit anti-tricellulin (1:1000, Invitrogen, Karlsruhe, Germany), and mouse anti-ZO-1 (1:500, Invitrogen, Karlsruhe, Germany). Alexa Fluor 488 goat anti-rabbit and Alexa Fluor 594 goat anti-mouse (1:500 each, Molecular Probes MoBiTec, Göttingen, Germany) were used as secondary antibodies. 4′,6-Diamidino-2-phenylindole (DAPI, 1:1000) was used to stain nuclei. Fluorescence images were acquired by a confocal laser scanning microscope (LSM 780, Carl Zeiss, Jena, Germany).

### 4.7. Electrophysiological and Paracellular Flux Measurements

Electrophysiological features were measured in Ussing chambers as described before [53]. For measuring FD4 fluxes, 0.4 mM of pre-dialyzed FITC-dextran 4 kDa (TdB Consultancy, Sweden) in apical hemi-chamber and 0.4 mM unlabeled dextran 4 kDa (Serva, Heidelberg, Germany) in basolateral side were applied. The measurements were carried out at 520 nm using a spectrometer (Tecan Infinite M200, Tecan, Switzerland) on basolateral samples taken at 0, 30, 60, 90, and 120 min after addition from the unlabeled side.

### 4.8. Statistical Analysis

Data are expressed as mean values ± standard error of the mean (SEM). Statistical analyses were performed using a Student’s *t*-test for comparison between two groups or one-way ANOVA for comparing more than two groups (multiple testing). *p* < 0.05 was considered significant.

## Figures and Tables

**Figure 1 ijms-21-07824-f001:**
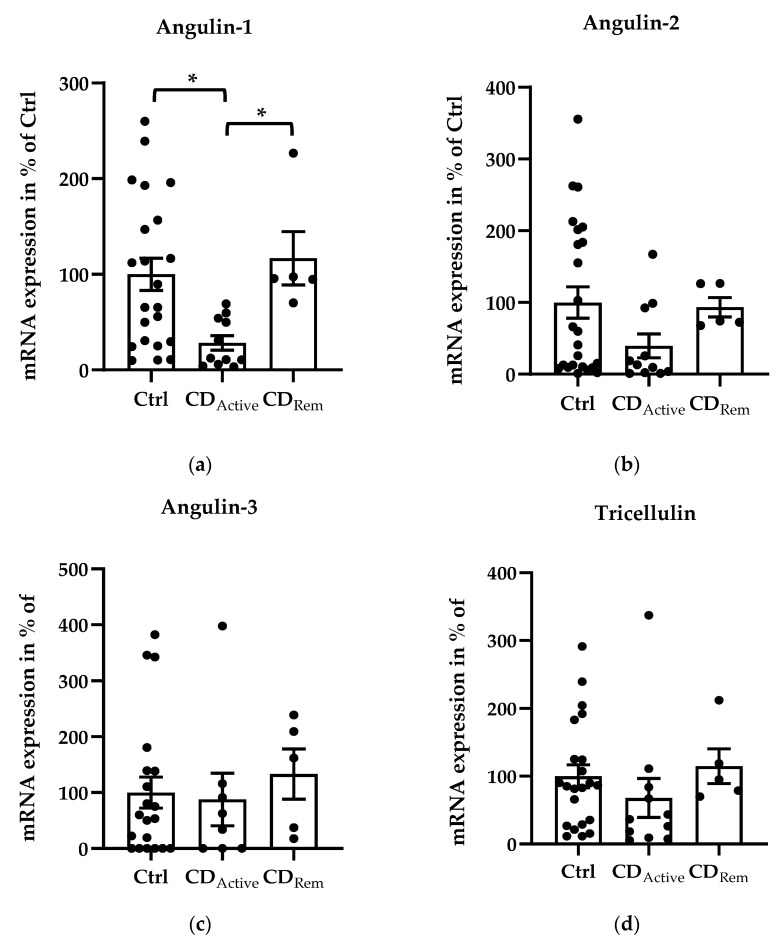
Scatter and bar plots of tight junction (TJ) protein mRNA expression analysis of human intestinal tissue. Mean value of controls (Ctrl) is set to 100%. (**a**) Angulin-1 is downregulated in active CD (28.26 ± 7.62%, *n* = 11) compared to Ctrl (100 ± 16.81%, *n* = 22, * *p* < 0.05) and remission patients (116.86 ± 27.87%, *n* = 5, * *p* < 0.05). (**b**) Angulin-2: Ctrl = 100 ± 21.85%, *n* = 24; CD_Active_ = 37.63 ± 16.14%, *n* = 11; CD_Rem_ = 97.44 ± 12.52%, *n* = 5). (**c**) Angulin-3: Ctrl = 100 ± 27.54%, *n* = 20; CD_Active_ = 87.78 ± 46.96%, *n* = 8; CD_Rem_ = 133.09 ± 44.91%, *n* = 5. (**d**) Tricellulin: Ctrl = 100 ± 16.75%, *n* = 22; CD_Active_ = 67.87 ± 28.82%, *n* = 11; CD_Rem_ = 114.90 ± 25.69%, *n* = 5. (**e**) Cldn-2 shows an increase in CDActive (256.28 ± 48.51%, *n* = 7) in comparison with Ctrl (100 ± 18.61%, *n* = 19, ** *p* < 0.01) and remission CD (76.67 ± 30.36%, *n* = 4, * *p* < 0.05). (**f**) Cldn-4: Ctrl = 100 ± 28.64%, *n* = 20; CD_Active_ = 127.67 ± 39.31%, *n* = 11; CD_Rem_ = 68.33 ± 29.68%, *n* = 5.

**Figure 2 ijms-21-07824-f002:**
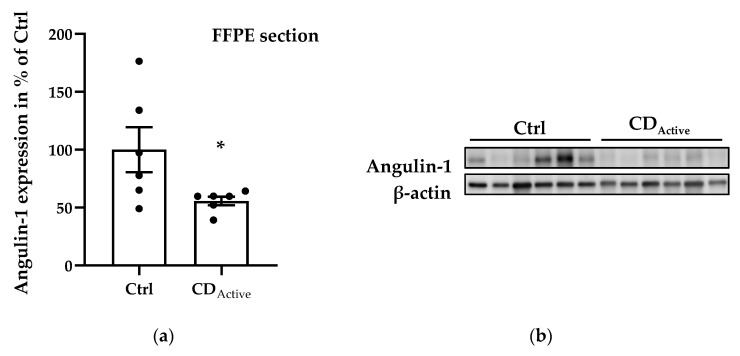
Angulin-1 protein expression analysis of human intestinal tissues. (**a**) Scatterplot with bar of angulin-1 in formalin-fixed paraffin-embedded (FFPE) section of Ctrl and CD. Mean value of Ctrl is set to 100%. Angulin-1 is downregulated in CD (Ctrl: 100 ± 19.42%, *n* = 6; CD: 55.82 ± 3.67%, *n* = 6, * *p* < 0.05). (**b**) Representative Western blots of intestinal tissues of Ctrl and CD. (**c**) Scatterplot with bar of angulin-1 in biopsies directly frozen after colonoscopy from Ctrl, active CD, and remission CD. Mean value of Ctrl is set to 100%. Angulin-1 is downregulated in active CD (CD_Active_: 55.59 ± 10.50%, *n* = 13) compared with Ctrl (Ctrl: 100 ± 7.95%, *n* = 15, ** *p* < 0.01) and remission CD (CD_Rem_: 120.00 ± 7.54%, *n* = 5, ** *p* < 0.01). (**d**) Representative Western blots of intestinal tissues of Ctrl and CD.

**Figure 3 ijms-21-07824-f003:**
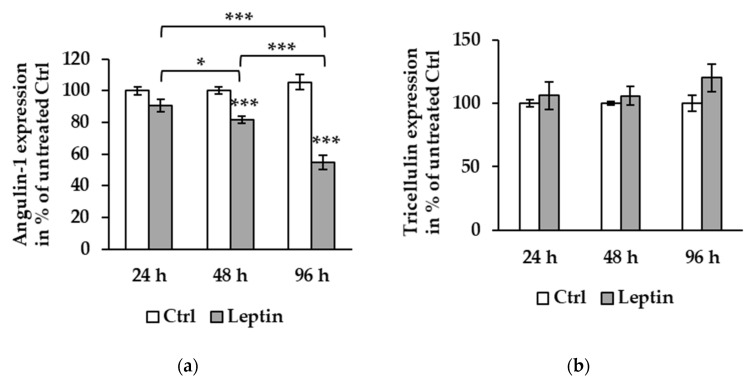
Protein expression effect of leptin at different time points in T84 cells. (**a**) Densitometric analysis shows angulin-1 expression to be 90.68 ± 3.83% at 24 h (*n* = 6), 81.11 ± 2.07% at 48 h (*** *p* < 0.001, *n* = 12), and 58.66 ± 2.62% at 96 h (*** *p* < 0.001, *n* = 12) compared to untreated Ctrl. The protein expression at 96 h is decreased compared to both 24 (*** *p* < 0.001) and 48 h (*** *p* < 0.001). At 48 h, angulin-1 expression is also lower than 24 h (* *p* < 0.05). (**b**) Tricellulin expression level shows no differences between 24 (*n* = 6), 48 (*n* = 12), and 96 h (*n* = 12).

**Figure 4 ijms-21-07824-f004:**
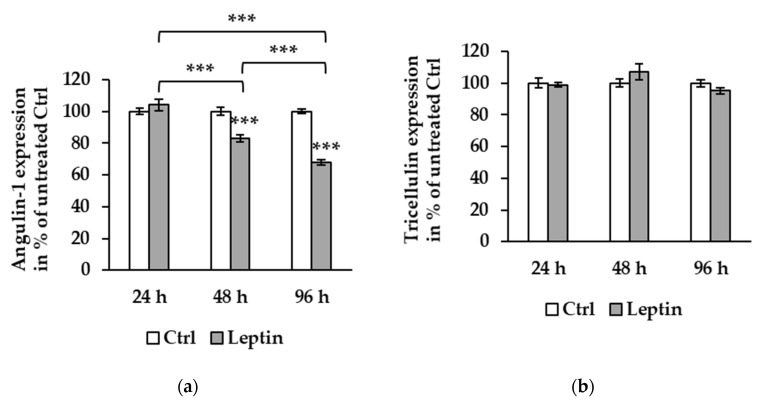
Effect of leptin on protein expression at different time points in Caco-2 cells. (**a**) Angulin-1 expression level exhibits reduction at 48 (82.38 ± 1.03%, *** *p* < 0.001, *n* = 12) and 96 h (67.71 ± 1.68%, *** *p* < 0.001, *n* = 12) of leptin treatment compared to untreated Ctrl. There is a significant difference between 24 and 48 h (*** *p* < 0.001), 48 (*** *p* < 0.001) and 96 h (*** *p* < 0.001), and 24 and 96 h. (**b**) Tricellulin expression is not affected by leptin at 24, 48, or 96 h.

**Figure 5 ijms-21-07824-f005:**
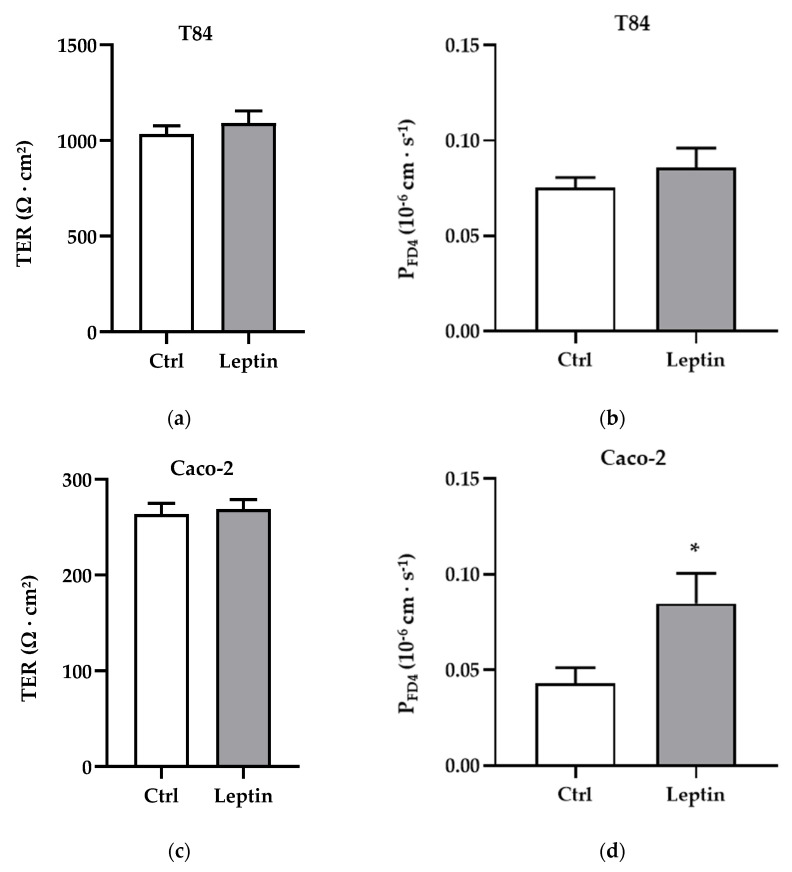
Functional analysis in T84 and Caco-2 cells. (**a**) Transepithelial resistance (TER) in T84 cells. (**b**) Permeability for FITC-dextran 4 kDa (P_FD4_) in T84 cells: Ctrl = 0.075 ± 0.005 × 10^−6^ cm·s^−1^, *n* = 9; leptin = 0.086 ± 0.010 × 10^−6^ cm·s^−1^, *n* = 8. (**c**) Transepithelial resistance in T84 cells. (**d**) Permeability for FITC-dextran 4 kDa in Caco-2 cells: Ctrl = 0.045 ± 0.008 × 10^−6^ cm·s^−1^, *n* = 9; leptin = 0.083 ± 0.016 × 10^−6^ cm·s^−1^, *n* = 11, * *p* < 0.05.

**Figure 6 ijms-21-07824-f006:**
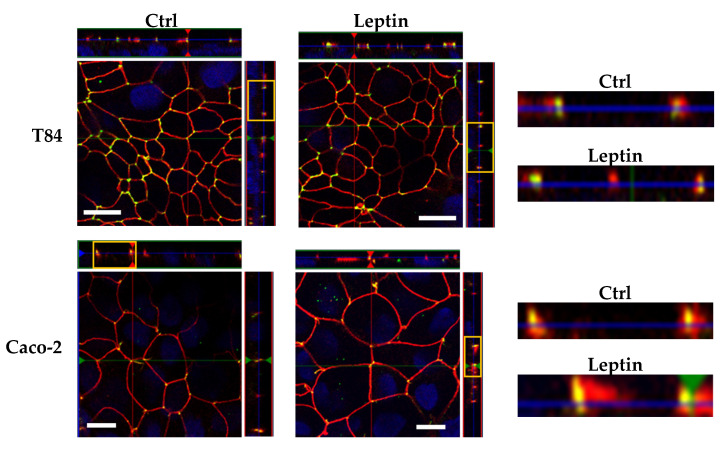
Representative immunofluorescent staining of tricellulin (green) and ZO-1 (red) in human intestinal cell lines. Nuclei were stained using 4’,6-diamidino-2-phenylindole (DAPI) (blue). With the TJ indicator ZO-1, the localization of tricellulin at tricellular TJ (tTJs) are demonstrated. The selected Z-scans (yellow box) are magnified on the right, indicating that there were no localization shifts in the lateral direction. Bars = 10 μm.

**Figure 7 ijms-21-07824-f007:**
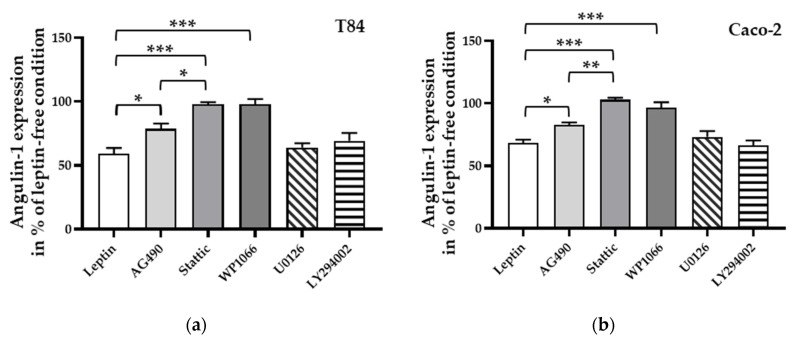
Densitometric analysis of angulin-1 expression after incubation with leptin and pre-treated with different inhibitors in T84 cells (**a**) and in Caco-2 cells (**b**). * *p* < 0.05, ** *p* < 0.01, *** *p* < 0.001, *n* = 6.

**Figure 8 ijms-21-07824-f008:**
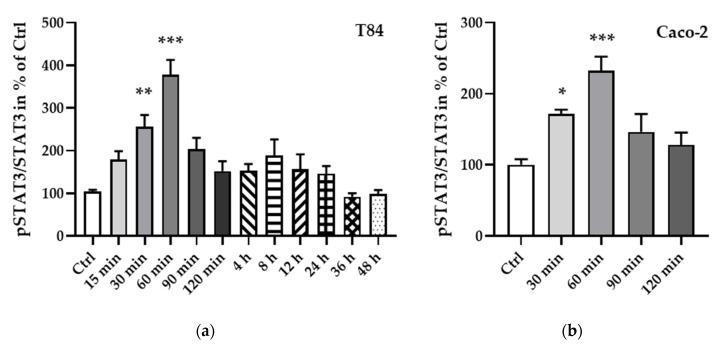
Phosphorylated STAT3 to total STAT3 ratio at different time points of leptin treatment in T84 (**a**) and Caco-2 cells (**b**). The phosphorylation of STAT3 becomes significant after 30 min of leptin treatment and peaks at 60 min in both cell lines. * *p* < 0.05, ** *p* < 0.01, *** *p* < 0.001, *n* = 6–11.

**Figure 9 ijms-21-07824-f009:**
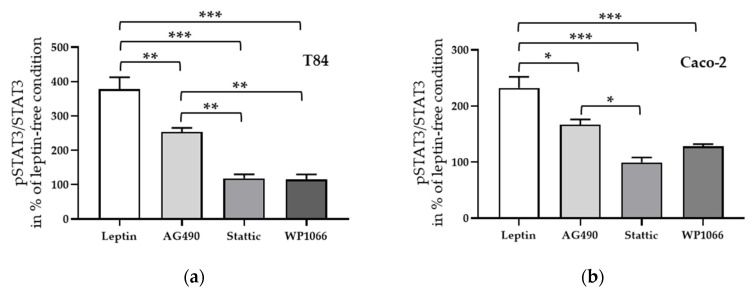
Effect of inhibitors on phosphorylated STAT3 to total STAT3 ratio in T84 (**a**) and in Caco-2 cells (**b**). * *p* < 0.05, ** *p* < 0.01, *** *p* < 0.001, *n* = 6–8.

**Table 1 ijms-21-07824-t001:** Characteristics of the enrolled Crohn’s disease (CD) population.

Characteristic	Controls (*n* = 24)	CD (*n* = 19)
Age (median, range)	54 (24–66)	35 (25–64)
Gender (male/female)	8/16	5/14
SES-CD, *n*		
Remission (0–2)	-	5
Active (> 2)	-	14

**Table 2 ijms-21-07824-t002:** Complete growth mediums for different cell lines.

Cell Line	Basic Medium	Source	Supplements
T84	DMEM: F-12 medium	Sigma-Aldrich, Steinheim, Germany	10% FBS, 100 U/mL penicillin, 100 μg/mL streptomycin
Caco-2	MEM with glutamax	Sigma-Aldrich, Steinheim, Germany	15% FBS, 100 U/mL penicillin, 100 μg/mL streptomycin

**Table 3 ijms-21-07824-t003:** Cytokines and inhibitors.

	Concentration	Source
**Cytokines**		
TNFα	500 U/mL	PeproTech, Hamburg, Germany
IFNγ	1000 U/mL	PeproTech, Hamburg, Germany
Leptin	200 ng/mL	PeproTech, Hamburg, Germany
IL-1β	100 ng/mL	PeproTech, Hamburg, Germany
IL-6	10 ng/mL	Miltenyi Biotec, Bergisch Gladbach, Germany
IL-12	100 ng/mL	PeproTech, Hamburg, Germany
IL-17A	100 ng/mL	PeproTech, Hamburg, Germany
IL-17F	100 ng/mL	Miltenyi Biotec, Bergisch Gladbach, Germany
IL-21	100 ng/mL	Miltenyi Biotec, Bergisch Gladbach, Germany
IL-22	50 ng/mL	PeproTech, Hamburg, Germany
IL-23	100 ng/mL	PeproTech, Hamburg, Germany
IL-33	100 ng/mL	Miltenyi Biotec, Bergisch Gladbach, Germany
**Inhibitors**		
AG490	100 μM	Calbiochem, Darmstadt, Germany
LY294002	10 μM	Calbiochem, Darmstadt, Germany
Stattic	20 μM	Calbiochem, Darmstadt, Germany
U0126	10 μM	Cell signaling Technology, Frankfurt am Main, Germany
WP1066	5 μM	Calbiochem, Darmstadt, Germany

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
