# Peer review of "Leptin Downregulates Angulin-1 in Active Crohn’s Disease via STAT3"

_ijms, 2020, doi:10.3390/ijms21217824_

Round 1
Reviewer 1 Report
The authors investigated the role of angulin-1 in Crohn’s disease, which is defined as a chronic inflammatory disease of the digestive tract. They observed a downregulation of this protein in active disease, compared to controls and patients in remission. To investigate a link between low expression of angulin-1 and Crohn’s disease, they used two cell lines, which are both described by ATCC as originated from adenocarcinoma of the colon. Furthermore, Caco-2 are immortalised cells. I do not consider that adenocarcinoma cells represent the signalling and interaction of colon epithelial cells, which were the target of this study. In addition, the data presentation has to be modified.
- The title should read “Leptin downregulates angulin-1 in active Crohn’s disease via STAT3”.
- Line 53: It is not clear what is the consequence of “the entry of pathogens and immunogens from the gut lumen”? What is the result of such barrier crossing for Crohn’s disease?
- Line 70: Do the authors assume that tricellulin is first expressed in the so called crypt of the small intestine epithelial cells, and then shifted to the villus structure? it is not clear how this mechanism should work.
- Figures 2A and C: Please describe how the expression of angulin-1 was quantified in the tissue sections. There is no such description in the methods.
- The data described in lines 131-137 suggest a cell line-specific response to Leptin. Since the effect of TNF-α and IL-17F was significant, it should not be described as “slightly increased”.
- The 96 hour-effect of Leptin lacks description of details. Was Leptin given once, or repeatedly over the 96 hours.
- The data on tricellulin presented in figures 3B and 4B indicate that angulin-1 has no effect on tricellulin, which should be described in the discussion more clearly. This result contradicts the link between the two proteins and their contribution to tight junction of epithelial cells in Crohn’s disease.
- Figure 8: If the kinetic of STAT3 activation is compared in both cell lines, the time scale of the two cell lines should be the same.
Author Response
Dear Editors and Reviewers,
thank you for your letter and for the reviewer's comments concerning our manuscript “Leptin induces angulin-1 downregulation in active Crohn's disease via STAT3”.
We have studied these comments carefully and tried our best to revise and improve the manuscript. The main corrections in the paper and the point-by-point responses to the reviewer's comments are as following:
Reviewer 1
General comment: The authors investigated the role of angulin-1 in Crohn’s disease, which is defined as a chronic inflammatory disease of the digestive tract. They observed a downregulation of this protein in active disease, compared to controls and patients in remission. To investigate a link between low expression of angulin-1 and Crohn’s disease, they used two cell lines, which are both described by ATCC as originated from adenocarcinoma of the colon. Furthermore, Caco-2 are immortalised cells. I do not consider that adenocarcinoma cells represent the signalling and interaction of colon epithelial cells, which were the target of this study.
Response: Thanks for raising this important point of representative cell lines. T84 and Caco-2 cells are both common cell lines used in various studies to represent colon epithelial cells and to elucidate signaling pathways as well as barrier function. Therefore, we believe that it is appropriate to perform in vitro experiments using these two lines in our study. Also, the similar results acquired from these two cell lines exclude cell-line-specific effect to some extent.
Comment 1: The title should read “Leptin downregulates angulin-1 in active Crohn’s disease via STAT3”.
Response: Good idea; we have changed the title accordingly.
Comment 2: Line 53: It is not clear what is the consequence of “the entry of pathogens and immunogens from the gut lumen”? What is the result of such barrier crossing for Crohn’s disease?
Response: Thank you for pointing this out. To explain the consequences of pathogen uptake we have added a short paragraph: "Under normal conditions, the intestinal epithelial barrier prevents significant uptake of luminal antigens into the Lamina propria. However, if the tTJ barrier is opened for the passage of large molecules, luminal antigens can enter the Lamina propria. There, they stimulate local immune cells to develop a pro-inflammatory response by releasing chemokines which exacerbate the intestinal inflammatory process."
Comment 3: Line 70: Do the authors assume that tricellulin is first expressed in the so called crypt of the small intestine epithelial cells, and then shifted to the villus structure? it is not clear how this mechanism should work.
Response: In the cited paper, the authors showed the immunofluorescent staining of colon biopsies from CD and controls: in controls, tricellulin was expressed along the crypt as well as the surface epithelium while in CD, tricellulin seemed to be reduced in the crypt but increased along the surface epithelium. These findings indicated a shifted localization of the expressed tricellulin in CD. Therefore, we expressed as “Expression of tricellulin was found to be shifted from depths of crypts to surface epithelium in CD”.
Comment 4: Figures 2A and C: Please describe how the expression of angulin-1 was quantified in the tissue sections. There is no such description in the methods.
Response: Thanks for the suggestion. We added the detailed description in the methods part. For samples from FFPE sections (Figure 2a), the protein was extracted according to the manufacturer’s instruction. For freshly taken biopsies (Figure 2c), the protein was extracted according to the description of cited paper. After extraction, the protein concentration was determined using BCA assay and based on the measured concentration, equal protein was loaded for Western blotting. The expression of angulin-1 in Figure 2A and 2C was quantified by comparing with the expression of β-actin which was a commonly used and rarely changed housekeeping protein for quantification.
Comment 5: The data described in lines 131-137 suggest a cell line-specific response to Leptin. Since the effect of TNF-α and IL-17F was significant, it should not be described as “slightly increased”.
Response: We deleted the word “slightly”.
Comment 6: The 96 hour-effect of Leptin lacks description of details. Was Leptin given once, or repeatedly over the 96 hours.
Response: Thanks for pointing out. We added the detail as “For 96 h of leptin treatment with or without inhibitors, the medium was renewed after 48 h along with the same concentration of leptin and corresponding inhibitors”.
Comment 7: The data on tricellulin presented in figures 3B and 4B indicate that angulin-1 has no effect on tricellulin, which should be described in the discussion more clearly. This result contradicts the link between the two proteins and their contribution to tight junction of epithelial cells in Crohn’s disease.
Response: The data in Figure 3 and 4 show that leptin downregulates angulin-1 but does not change the expression of tricellulin, which fits to the results from biopsies that in CD patients, angulin-1 is downregulated while tricellulin expression remains unchanged compared with controls. These results do not contradict the link between the two proteins because angulin-1 is known to be responsible to the correct localization of tricellulin but not for regulating its expression. Furthermore, the increase of permeability for FD4 in Caco-2 cells indicates that downregulation of angulin-1 indeed has an influence on barrier function.
Comment 8: Figure 8: If the kinetic of STAT3 activation is compared in both cell lines, the time scale of the two cell lines should be the same.
Response: Thanks for the suggestion. However, as we do not compare T84 cells and Caco-2 cells with each other, so after analyzing a more complete time scale in T84 cells, we think it is reasonable to investigate a relative narrow range of time scale in Caco-2 cells, especially when the results turn out to be similar in both cell lines.
We look forward to hearing from you regarding our submission and hope the revised version is now suitable for publication.
Sincerely,
Jia-Chen E. Hu, Michael Fromm, Susanne Krug
Reviewer 2 Report
The manuscript is of some interest, however, I suggest it would. Be improved by including a disease control group such as ulcerative colitis.
Author Response
Dear Editors and Reviewers,
thank you for your letter and for the reviewer's comments concerning our manuscript “Leptin induces angulin-1 downregulation in active Crohn's disease via STAT3”.
We have studied these comments carefully and tried our best to revise and improve the manuscript. The main corrections in the paper and the point-by-point responses to the reviewer's comments are as following:
Reviewer 2
Comment: The manuscript is of some interest, however, I suggest it would be improved by including a disease control group such as ulcerative colitis.
Response: Thanks for this suggestion. It would have been interesting to explore this aspect. However, our study is based on (i) tricellulin showed an unchanged expression level but a shifted localization in CD and (ii) angulins were responsible for the correct localization of tricellulin. Therefore, we here have analyzed the change of angulins in CD. Regarding UC, as tricellulin was shown to be downregulated, it would be of limited significance to compare these two diseases in this context and might lead to more confusion. In addition, to obtain a reasonable number of proper human biopsies takes a long time and cannot be done within a few months.
We look forward to hearing from you regarding our submission and hope the revised version is now suitable for publication.
Sincerely,
Jia-Chen E. Hu, Michael Fromm, Susanne Krug
Round 2
Reviewer 1 Report
no more changes required
Author Response
Thank you for accepting the changes we made.
Reviewer 2 Report
none
Author Response

(The authors gave the same response as above.)
